# A primary school-based dengue solution model for post-COVID-19 in southern Thailand: Students understanding of the dengue solution and larval indices surveillance system

Jiraporn Jaroenpool[1,2], Sarunya Maneerattanasak[3], Femi Adesina[4], Atchara Phumee[1,2], Muhammad Haroon Stanikzai[2,5,6], Chumpon Ponprasert[7], Yingqin Zheng[8], Shamarina Shohaimi[9], Truong Thanh Nam[10], Temesgen Anjulo Ageru[2,5], Amin Nordin Syafinaz[11], Charuai Suwanbamrung[2,5]*

1 School of Allied Health Sciences, Walailak University, Nakhon Si Thammarat, Thailand, 2 Excellent Center for Dengue and Community Public Health (EC for DACH), Walailak University, Nakhon Si Thammarat, Thailand, 3 Faculty of Medicine Siriraj Hospital, Department of Microbiology, Mahidol University, Bangkok, Thailand, 4 Department of Biology, Federal University of Technology Akure, Akure, Ondo State, Nigeria, 5 Public Health Research Program, School of Public Health, Walailak University, Nakhon Si Thamarata, Thailand, 6 Faculty of Medicine, Department of Public Health, Kandahar University, Kandahar, Afghanistan, 7 Lansaka District Public Health Official, Nakhon Si Thammarat Province, Thailand, 8 Essex Business School, University of Essex, Colchester, United Kingdom, 9 Faculty of Science, Department of Biology, Universiti Putra Malaysia, Serdang, Malaysia, 10 Faculty of Public Health, Can Tho University of Medicine and Pharmacy, Can Tho, Vietnam, 11 Faculty of Medicine and Health Sciences, Department of Medical Microbiology, Universiti Putra Malaysia, Serdang, Selangor, Malaysia

* yincharuai@gmail.com

## Abstract

One of the consequences of the COVID-19 lockdown is that it hinders school-based dengue management interventions. This is due to the closure of schools and the limited availability of online lessons in certain schools. Conversely, the level of basic understanding that primary school children have about the condition is directly related to their likelihood of getting it and their ability to modify their behaviour to prevent it. Thus, the study aims to assess the understanding and develop a school-based model intervention for dengue solutions and larval indices surveillance systems suitable for primary schools. The study used a purposive and convenience technique to recruit participants in the Keawsan sub-district in Nabon district, Nakhon Si Thammarat province, Thailand. An applied community participatory action research (CPAR) design was used to develop a school-based model intervention. The pre-test result shows that more than 90% and 100% of the students have a poor understanding of dengue solutions (UDS) and larval indices surveillance systems (ULISS), respectively. The students with good scores on UDS increased from 2.9% to 54.6% after the intervention, and there is still a prevalent lack of good understanding among more than 85% of the students on ULISS. Meanwhile, gender, class, and school significantly influence (p <0.05) UDS and ULISS among students in either the pre-test or post-test. Based on the thematic analysis, it was determined that the students had learned about dengue fever transmission,

**Data Availability Statement:** The authors confirm that all data underlying the findings are fully available without restriction. All relevant data are within the paper and its Supporting Information files.

**Funding:** This study was financially supported by the School of Public Health and Excellent Center for Dengue and Community Public Health, Walailak University. The funders had no role in study design, data collection and analysis, decision to publish, or preparation of the manuscript.

**Competing interests:** The authors declare no conflict of interest.

dengue mosquitoes, container index calculation, dengue symptoms, and how to prevent dengue mosquitoes, among other topics. Overall, it was discovered that the trained students could convey their knowledge of the dengue solution and the larval indices surveillance system to other students. We recommend that further ULISS training be provided for schoolchildren in more simple terms that they can understand.

## Introduction

Southeast Asian countries are experiencing a rise in dengue fever cases, which has become an additional issue in many of these countries as the world works to stop the COVID-19 virus from spreading. Dengue is an infectious disease transmitted mostly by the *Aedes aegypti* mosquito and, to a lesser extent, by *Aedes albopictus*. The dengue virus (DENV), which is the source of this infection, is an RNA virus from the family Flaviviridae and genus Flavivirus, with four serotypes (DENV1, DENV2, DENV3, and DENV4) [1]. Dengue fever is rapidly spreading throughout Southeast Asia, to the extent that between 2015 and 2019, dengue cases increased by 46% [2]. The COVID-19 pandemic and the dengue outbreak have resulted in a double burden of outbreaks in ASEAN countries, with dengue still a problem during the 2019–2020 outbreak [3]. The movement of individuals over national borders is the main factor contributing to the spread of both COVID-19 and dengue infection, which increases the likelihood of a continuous outbreak [4,5].

In 2020, dengue incidence was historically low in most nations due to disruptions caused by COVID-19. The COVID-19 pandemic guidelines are effective in preventing the spread of dengue both within and between urban and suburban areas, as well as in lowering the co-circulation of different genotypes or serotypes. To reduce COVID-19 transmission, public health measures like restricting movement and closing schools, universities, and offices resulted in an unanticipated significant fall in the reported number of dengue cases in Sri Lanka. Due to travel restrictions, the lower prevalence of dengue in Sri Lanka is believed to have been caused by less availability of blood meals and decreased vector density, especially of *Aedes aegypti* [6]. Contrarily, during the COVID-19 pandemic, dengue cases per million rose throughout Peru and in several endemic locations, with the exception of Piura [7]. Meanwhile, there was no significant impact on dengue incidence in Malaysia or Singapore [8]. This suggests no proof of a rise or fall in dengue incidence associated with adopting COVID-19 preventative strategies. Dengue epidemiology could be significantly impacted by the disparities in geographic characteristics amongst the nations mentioned above. From the spatial analysis, Sheng et al. [9] opined that after the dengue outbreak evolved, its distribution was relatively stable, and even resistant to COVID-19 measures, emphasising the possibility of developing an accurate prediction technique.

The earliest reports of DENV3 and DENV4 in Thailand and the Philippines occurred in 1953 [1]. Thailand has recorded more cases in 2024 than during the same period in 2023. Since the start of 2024, the Thailand Bureau of Epidemiology has recorded 26,527 dengue cases overall, which is a 123 percent increase over the 11,885 cases reported during the same period in 2023 [10,11]. Dengue exerts a substantial adverse macroeconomic effect on Thailand, diminishing GDP and resulting in losses for individuals and enterprises. Thailand's GDP was lowered by more than 0.33% (1.81 billion USD) as a result of it, along with employee income and expenditure declines, lost tourism earnings, and an increase in the direct medical costs of hospital and outpatient treatment [12].

The Keawsan subdistrict, which includes ten villages, 2,372 households, 7,068 people, three temples, four children's development centres, two primary care units, 134 village health volunteers (VHVs), a high school, and four primary schools, has been designated as a dengue risk area in southern Thailand. Before the COVID-19 pandemic, dengue morbidity rates during the previous five years (2015–2019) were 201, 218, 267, 350, and 533 cases per 100,000 people, respectively. The morbidity rates were higher than the 50 cases per 100,000 population threshold set by the Thai Ministry of Public Health. In particular, among children under 15, this area is still considered a high risk for dengue epidemics by the Department of Disease Control, Provincial Health Office, Nakhon Si Thammarat. All four primary schools were closed during the COVID-19 pandemic, and some students attended classes virtually. No dengue prevention initiatives were taking place then. In 2022, the school reopened for regular classes and followed the implementation of the COVID-19 preventive guidelines. According to the Keawsan subdistrict administrative organisation, the morbidity of dengue was reported to have increased as a consequence of students being infected with the virus at school after it reopened. While there have been no fatalities resulting from dengue illness, it is necessary to implement a targeted program for primary schools in the area. It is crucial to continuously monitor the occurrence of dengue as the COVID-19-associated restrictions are eased. This monitoring can provide valuable information about the transmission mechanisms and potential interventions for dengue [13]. Lockdown methods have presented opportunities to enhance dengue vector control efforts at the household, neighbourhood, and social levels. It is crucial to uphold efficient mosquito control methods [3].

In 2022, Nontapet et al. discovered that students in Keawsan have misconceptions regarding dengue, and the schools lack clear activities and networking relationships with neighbouring schools and the surrounding environment about dengue control. The COVID-19 epidemic may have diminished student awareness and hampered school-based dengue prevention efforts. Lockdowns significantly reduced the efficacy of vector control, particularly with regard to a lack of social mobilisation efforts and indoor and peridomestic pesticide spraying in private areas [14]. A review of research on dengue prevention in schools indicated that elementary school children's fundamental knowledge about the disease correlates to their risk of incidence and behavioural changes for prevention [15,16]. Developing prevention models and incorporating stakeholders in dengue education improved students' knowledge [17]. This discovery aligned with a study conducted on the behaviour and knowledge of fifty elementary school students, which demonstrated the significance of practical learning for these learners [18].

Numerous models have been established on the transmission dynamics, the Wolbachia mosquito replacement, and so on [19,20]. However, a school-based dengue solution model places particular emphasis on the age group (schoolchildren) who are most exposed to the disease. This model is vital because this age group can easily comprehend it, and it will not only lower larval indices on school premises but also in the homes of the students. Since the larval indices surveillance system is one of the most important preventive measures, it is an essential part of this approach. There is limited research that concentrates on understanding dengue solution (UDS) and larval indices surveillance systems (ULISS), and after COVID-19, it is important to update the previous dengue solution strategies to combat dengue in Thailand. There is an immediate need for an innovative dengue solution model in addition to the present model given the current increase in dengue infection cases in Thailand compared to the previous year.

To address this issue, the Kaewsan Subdistrict Administration Organization collaborated with the Quality Centre for Dengue, Community Health Networks, and the four elementary schools to develop a larval indices surveillance system model. Through a school-based

approach, the initiative focuses on improving understanding of the system and dengue fever solutions in light of the COVID-19 pandemic. Therefore, the objectives of the study were to: 1) assess the understanding of dengue solutions and larval indices surveillance systems among students and 2) develop a school-based model intervention for dengue solutions and larval indices surveillance systems that is suitable for primary schools.

## Materials and methods

### Study setting and participants

The four primary schools in Keawsan subdistrict in Nabon district, Nakhon Si Thammarat province, Thailand, are related to other stakeholders such as 13 villages, two primary care units, a subdistrict administration organisation, and village health volunteers (VHVs). The target population was in elementary 4, 5, and 6 of each school, then students who participated based on the availability of a leader and general students (Table 1). The leader students were delegates who represented all other students in participating, taking activities to prevent and control dengue problems, and motivating other students to be involved in the activities.

The purposive technique for all participants in the project meeting was based on the inclusion criteria; they were essential stakeholders or individuals with critical information about dengue solutions and a willingness to participate. The convenience technique for all student participants was based on their willingness and availability. The student participants were permitted to willingly choose which of the two groups (leader and non-leader groups) they wanted to join. Participants were enrolled and informed of the research objectives; their informed written or oral consent was obtained prior to participation. In particular, parental approval was secured to allow their children to participate in the experiment.

### Development step

To develop a school-based dengue model that was appropriate for the context, an applied community participatory action research (CPAR) design was used, comprising four steps: 1) preparation, 2) plan, 3) implementation, and 4) evaluation.

### Preparation step

The research team collaborated with the Keawsan SAO's leader and committee to prepare for this study. The project meeting was scheduled with the SAO committee, which included two SAO representatives, a district health officer, eight teachers and leaders from four schools,

Table 1. Number of students in grades 4, 5, and 6 in each school and number of buildings.

| Primary School | Number of buildings in the school | The service area for students (village) | Total students in school (students) | Number of the target population (students) | | | |
|---|---|---|---|---|---|---|---|
| | | | | Elementary 4 | Elementary 5 | Elementary 6 | Total |
| A | 3 (2–3 floors) | 5 | 202 | 46 | 32 | 44 | 122 |
| B | 4 (one floor) | 2 | 90 | 12 | 19 | 31 | 62 |
| C | 2 (one floor) | 1 | 129 | 27 | 31 | 18 | 76 |
| D | 5 (small and one floor) | 5 | 113 | 17 | 32 | 17 | 66 |
| Four schools | | 13 villages | 534 | 102 | 114 | 110 | 326 |

health professional officials, and community leaders. The purpose of the meeting was to outline the project's aims and activities to solve the dengue problem, as well as the schedule and roles of all stakeholders in the schools.

1. Set the response group of the student leaders to "Club." Practical training sessions were held in each school to install and improve the competencies of the leadership teams. The training consisted of examining the understanding of dengue illness, offering instruction, and teaching the processes. There is one "Mosquito Control and Dengue Prevention Club" at each school.

2. The mosquito control and dengue prevention clubs carried out activities to support the surveillance system activities of each school. The role of the student leader is to promote and encourage activities for dengue prevention and control in their school.

3. Leader student's UDS and ULISS Assessment

Bloom's updated taxonomy for learning categorises "understanding" as one of the six cognitive levels: knowledge, comprehension (understanding), application, analysis, synthesis, and assessment. In this study, the knowledge of the dengue solution (UDS) refers to the capacity of the students to comprehend dengue prevention, control, and self-care. Meanwhile, understanding the larval indices surveillance system (ULISS) relates to the student's competence in learning the larval indices that characterise the larval indices surveillance system operations and the larval index levels.

Plan of school-based activities

1. Meeting the school's stakeholders to plan for dengue solutions.

2. Promoting dengue solutions in front of the flagpole.

3. Setting up a dengue information board.

4. Drawing picture-related dengue solutions.

5. Painting a picture-related dengue solution.

6. Holding a dengue slogan contest.

7. Organising a walking campaign to raise awareness about dengue solutions.

8. Showing a drama play to reflect dengue solutions.

## Implementation step

The activities were conducted for 12 weeks, comprising the UDS and ULISS training programs and the designation of responsibilities for each stakeholder. Education training related to dengue solution and larval indices surveillance system in primary school. The coordinator of the training made sure that the study materials were taught in depth in order to complete the questionnaire.

## Evaluation step

This step consists of evaluating the development process and the intervention model's effects. We measured and compared the UDS and ULISS of leaders and non-leader students before and after using the model.

## Questionnaires for assessment and evaluation

Before designing the questionnaire, the stakeholders and the research team conducted a general evaluation of the study contents for the training program to meet the research objectives. This study's contents were developed by evaluating reports and papers on dengue infection and instructional methods. The initial draft of a questionnaire was examined and analysed by researchers, language experts, and questionnaire analysts. The questionnaire was in both English and Thai. The researchers revised and resubmitted the questionnaire in response to the expert assessment feedback.

The non-leader and leader students' UDS and ULISS were examined using the same questionnaires, which had been adjusted to include 10 UDS and 10 ULISS items. Three experts evaluated the CVIs and found values of 0.82 for the UDS items and 0.88 for the ULISS items, confirming their validity. Cronbach alpha coefficients for the UDS and ULISS of leader students were 0.80 and 0.81, respectively, while for non-leader students, the Cronbach alpha coefficients for the UDS and ULISS were 0.72 and 0.73, respectively; these values demonstrated acceptable reliability (alpha) of 0.70 or higher [21].

We calculated mean scores for both the UDS and the ULISS to characterise good and poor levels of understanding, based on Bloom's cut-off score of 80% [22,23]. More specifically, answering correctly on 80% or more of the questions showed a good understanding; answering incorrectly on less than 80% indicated a poor understanding.

The teachers took the student leaders on a physical survey of the school's buildings and surroundings, looking for items like drinking water containers, tires with water, water containers in the restroom and kitchen, vases, cupboard saucers in the cafeteria, containers containing plants, stagnant water, and discarded containers in and around the building. The students' leader had previously been taught how to survey mosquito larvae and calculate larval indices in class. The survey methodology, mosquito breeding areas, how to identify larvae, and how to count, record, and calculate them were all demonstrated to the students' leaders. Students were requested to repeat the same process at home and at school on a regular basis and to report the survey results to the assigned classroom teacher, who subsequently forwarded them to the study team.

The procedure and results promoted regular dengue prevention and control measures in schools. The study team regularly inspects the area surrounding school buildings to look for mosquito breeding grounds, evaluate the success of the leadership program, and determine how well students are implementing what they have learned to eradicate mosquito vectors from the school.

The outcomes and feedbacks from the students (leader and non-leader groups) and teachers, as well as the assessment of the executions and analysis before and after the study team's activities, were gathered and presented at the subsequent meeting. During the re-evaluation stage, stakeholders met to discuss and reflect on the outcome. The participants reflected on components of the project, such as their participation in the activities, the larval indices surveillance system, the dengue training program, and the project's utilities, barriers, and success, and they proposed solutions to dengue. Students were given the dengue handbook materials from this session to take home and distribute to their family members, particularly their parents. Every school officially created a Dengue Prevention and Mosquito Control Club and invited additional students to join.

## Data analysis

The data on the participants' characteristics were collected from leader students and non-leader students and analysed by descriptive statistics. The chi-square test and Fisher's exact test

were used to compare the scores for the UDS and ULISS before and after the implementation of the program, and the results of the school-based model were shown in tables and figures.

The qualitative data from the group discussion meetings of the student leaders were evaluated by utilising Braun and Clarke's theme analysis [24,25]. Two researchers conducted and examined the reflections of the leader students, non-leader students, health instructors, and parents who were offered in meetings held at the closure of the project.

## Ethics approval

The research ethics were considered by the Human Research Ethics Committee of Walailak University, No. WUEC-22-223-0. Participants were informed of the research objectives; their informed written or oral consent was obtained prior to participation. In particular, parental approval was secured to allow their children to participate in the experiment.

## Result

Table 2 shows the recruited numbers of non-leaders and leaders from each school studied. There were a total of 105 non-leader and 58 leader students for this study.

Checking the experience of the students about dengue diseases (Table 3), for leader and non-leader students, there was no significant difference ($p > 0.05$) between the pre-test and post-test study between those who have had the experience of dengue infection in the past 12 months, from another student or their family members. Meanwhile, for leader students, there was a significant difference ($p < 0.05$) between the pre-test and post-test on experiencing dengue illness from the neighbourhood, but not statistically significant for non-leader students. The number of outcomes in the post-test varied from the pre-test study.

The understanding of dengue solutions was significantly different ($p < 0.05$) among the schools for the post-test (Table 4). Generally, the outcome of the good understanding of dengue solutions of the non-leader students increased in the post-test. Meanwhile, there was a significant difference ($p < 0.05$) in the post-test of leader and non-leader students among classes, and in the pre-test and post-test among schools for leader students. Also, the outcome of leader students' good understanding of dengue solutions increases in the post-test.

In terms of understanding the larval indices surveillance system, there was an absolutely poor understanding outcome for the non-leader and leader pre-tests (Table 5). For non-leader students' post-test, there was a significant difference ($p < 0.05$) among schools in their understanding of the larval indices surveillance system. For leader students' post-test, there were

**Table 2. Students' information using a school-based model.**

|  |  |  | A | B | C | D | Total |
|---|---|---|---|---|---|---|---|
| Gender | Female | Non-leader | 47 (45.2) | 19 (18.3) | 13 (12.5) | 25 (24.0) | 104 |
|  |  | Leader | 20 (27.0) | 11 (14.9) | 27 (36.5) | 16 (21.6) | 74 |
|  | Male | Non-leader | 41 (38.7) | 27 (25.5) | 19 (17.9) | 19 (17.9) | 106 |
|  |  | Leader | 14 (33.3) | 5 (11.9) | 17 (40.5) | 6 (14.3) | 42 |
| Class | Elementary 4 | Non-leader | 31 (50.8) | 7 (11.5) | 13 (21.3) | 10 (16.4) | 61 |
|  |  | Leader | 15 (36.6) | 5 (12.2) | 14 (34.1) | 7 (17.1) | 41 |
|  | Elementary 5 | Non-leader | 14 (24.6) | 13 (22.8) | 11 (19.3) | 19 (33.3) | 57 |
|  |  | Leader | 18 (31.6) | 6 (10.5) | 20 (35.1) | 13 (22.8) | 57 |
|  | Elementary 6 | Non-leader | 43 (46.7) | 26 (28.3) | 8 (8.7) | 15 (16.3) | 92 |
|  |  | Leader | 1 (5.6) | 5 (27.8) | 10 (55.6) | 2 (11.1) | 18 |

**Table 3. Comparison of student's experience with dengue illness before and after training.**

| | | Non-leader | | | | Leader | | | |
|---|---|---|---|---|---|---|---|---|---|
| | | Time | | $\chi^2$ | p-value | Time | | $\chi^2$ | p-value |
| | | pre-test | post-test | | | pre-test | post-test | | |
| Have experienced dengue illness in the past 12 months | No | 90 (50.6) | 88 (49.4) | 0.147 | 0.70 | 50 (50.0) | 50 (50.0) | 0 | 1 |
| | Yes | 15 (46.9) | 17 (53.1) | | | 8 (50.0) | 8 (50.0) | | |
| Have the experience of dengue illness by student | No | 90 (49.7) | 91 (50.3) | 0.04 | 0.84 | 55 (50.5) | 54 (49.5) | 0.152 | 0.69 |
| | Yes | 15 (51.7) | 14 (48.3) | | | 3 (42.9) | 4 (57.1) | | |
| Have the experience of dengue illness from neighbourhood | No | 78 (47.0) | 88 (53.0) | 2.875 | 0.09 | 42 (60.9) | 27 (39.1) | 8.048 | 0.01* |
| | Yes | 27 (61.4) | 17 (38.6) | | | 16 (34.0) | 31 (66.0) | | |
| Have the experience of dengue illness from a family member | No | 92 (48.4) | 98 (51.6) | 1.989 | 0.16 | 51 (49.5) | 52 (50.5) | 0.087 | 0.77 |
| | Yes | 13 (65.0) | 7 (35.0) | | | 7 (53.8) | 6 (46.2) | | |

$\chi^2$–Chi-Square.

*Significant at *p<0.05, Fisher's exact test was used.*

significant differences (p <0.05) among gender, classes, and schools in their understanding of the larval indices surveillance system.

Fig 1 shows that there was a poor understanding of dengue illness among the studied students (2.9–5.2%), and there were no significant differences (p > 0.05) between non-leaders and leaders in the pre-test study. Meanwhile, in the post-test, a good understanding of dengue illness increased among non-leaders from 2.9% to 33.3%. The number of leader students, on the other hand, increased from 5.2% to 54.6% in good understanding. So, a significant difference (p <0.05) exists between non-leader and leader students' post-test studies.

There were 100% non-leader and leader students who had a poor understanding of the larval indices surveillance system in the pre-test study (Fig 2). Meanwhile, the number of students with good understanding increased to 10.9% and 12.1%, respectively, in the post-test study, and there was no significant difference (p > 0.05) between the two groups of students. In Fig 3, the highest number of leaders that indicated larval indices surveyed behaviour at school

**Table 4. The association between characteristics and understanding of dengue solution (UDS) in pre-test and post-test between leader and non-leader students.**

| | | Non-leader | | | | | | Leader | | | | | |
|---|---|---|---|---|---|---|---|---|---|---|---|---|---|
| | | Pre-test | | P—value | Post-test | | P—value | Pre-test | | P—value | Post-test | | P—value |
| | | Poor | Good | | Poor | Good | | Poor | Good | | Poor | Good | |
| Gender | Female | 53 (96.4) | 2 (3.6) | 0.62 | 35 (63.6) | 20 (36.4) | 0.80 | 34 (91.9) | 3 (8.1) | 0.18 | 14 (37.8) | 23 (62.2) | 0.08 |
| | Male | 49 (98.0) | 1 (2.0) | | 33 (66.0) | 17 (34.0) | | 21 (100.0) | 0 (0.0) | | 13 (61.9) | 8 (38.1) | |
| Class | El 4 | 24 (100.0) | 0 (0.0) | 0.17 | 10 (41.7) | 14 (58.3) | 0.03* | 19 (95.0) | 1 (5.0) | 0.75 | 13 (65.0) | 7 (35.0) | 0.03* |
| | El 5 | 32 (100.0) | 0 (0.0) | | 21 (65.6) | 11 (34.4) | | 28 (93.3) | 2 (6.7) | | 8 (26.7) | 22 (73.3) | |
| | El 6 | 46 (93.9) | 3 (6.1) | | 36 (73.5) | 13 (26.5) | | 8 (100.0) | 0 (0.0) | | 4 (50.0) | 4 (50.0) | |
| School | A | 44 (100.0) | 0 (0.0) | 0.13 | 36 (81.8) | 8 (18.2) | 0.00** | 17 (100.0) | 0 (0.0) | 0.04* | 14 (82.4) | 3 (17.6) | 0.01* |
| | B | 23 (100.0) | 0 (0.0) | | 15 (65.2) | 8 (34.8) | | 8 (100.0) | 0 (0.0) | | 4 (50.0) | 4 (50.0) | |
| | C | 15 (93.8) | 1 (6.3) | | 5 (31.3) | 11 (68.8) | | 22 (100.0) | 0 (0.0) | | 5 (22.7) | 17 (77.3) | |
| | D | 20 (90.9) | 2 (9.1) | | 19 (86.4) | 3 (13.6) | | 8 (72.7) | 3 (27.3) | | 4 (36.4) | 7 (63.6) | |

$\chi^2$–Chi-Square.

*Significant at **p<0.001, *p<0.05, Fisher's exact test used.*

**Table 5. Associate between characteristics and understanding larval indices surveillance system (ULISS) of pre-test and post-test among leader and non-leader.**

| | | Non-leader | | | | | | Leader | | | | | |
| --- | --- | --- | --- | --- | --- | --- | --- | --- | --- | --- | --- | --- | --- |
| | | Pre-test | | P—value | Post-test | | P—value | Pre-test | | P—value | Post-test | | P—value |
| | | Poor | Good | | Poor | Good | | Poor | Good | | Poor | Good | |
| Gender | Girl | 55 (100.0) | 0 (0.0) | - | 52 (94.5) | 3 (5.5) | 0.14 | 37 (100.0) | 0 (0.0) | - | 35 (94.6) | 2 (5.4) | 0.04* |
| | Boy | 50 (100.0) | 0 (0.0) | | 43 (86.0) | 7 (14.0) | | 21 (100.0) | 0 (0.0) | | 16 (76.2) | 5 (23.8) | |
| Class | El 4 | 24 (100.0) | 0 (0.0) | - | 20 (83.3) | 4 (16.7) | 0.35 | 20 (100.0) | 0 (0.0) | - | 20 (100.0) | 0 (0.0) | 0.02* |
| | El 5 | 32 (100.0) | 0 (0.0) | | 29 (90.6) | 3 (9.4) | | 30 (100.0) | 0 (0.0) | | 23 (76.7) | 7 (23.3) | |
| | El 6 | 49 (100.0) | 0 (0.0) | | 46 (93.9) | 3 (6.1) | | 8 (100.0) | 0 (0.0) | | 8 (100.0) | 0 (0.0) | |
| School | A | 44 (100.0) | 0 (0.0) | - | 44 (100.0) | 0 (0.0) | 0.00* | 17 (100.0) | 0 (0.0) | - | 17 (100.0) | 0 (0.0) | 0.04* |
| | B | 23 (100.0) | 0 (0.0) | | 22 (95.7) | 1 (4.3) | | 8 (100.0) | 0 (0.0) | | 8 (100.0) | 0 (0.0) | |
| | C | 16 (100.0) | 0 (0.0) | | 7 (43.8) | 9 (56.3) | | 22 (100.0) | 0 (0.0) | | 16 (72.7) | 6 (27.3) | |
| | D | 22 (100.0) | 0 (0.0) | | 22 (100.0) | 0 (0.0) | | 11 (100.0) | 0 (0.0) | | 10 (90.9) | 1 (9.1) | |

$\chi^2$–Chi-Square.

*Significant at **$p<0.001$, $p<0.05$, Fisher's exact test used.

and home every seven days (77.6% and 89.7%, respectively). Meanwhile, the highest non-leader with larval indices surveyed behaviour in school and home was every seven days (58.1%).

Based on the provided responses, a thematic analysis revealed four key themes: knowledge acquisition, practical application, and awareness regarding dengue mosquitoes and fever (Table 6).

## Discussion

In Thailand, dengue fever has been spreading for over 60 years. However, there is no specific epidemic pattern due to the circulation of 4 dengue serotypes. Additionally, there are regular occurrences of epidemics in the central and southern regions of Thailand, as noted by Suwan-bamrung et al. [26]. Several authors have observed that the highest number of cases occurred in the age range of >11 years (34.02%), whereas newborns were the least affected age group

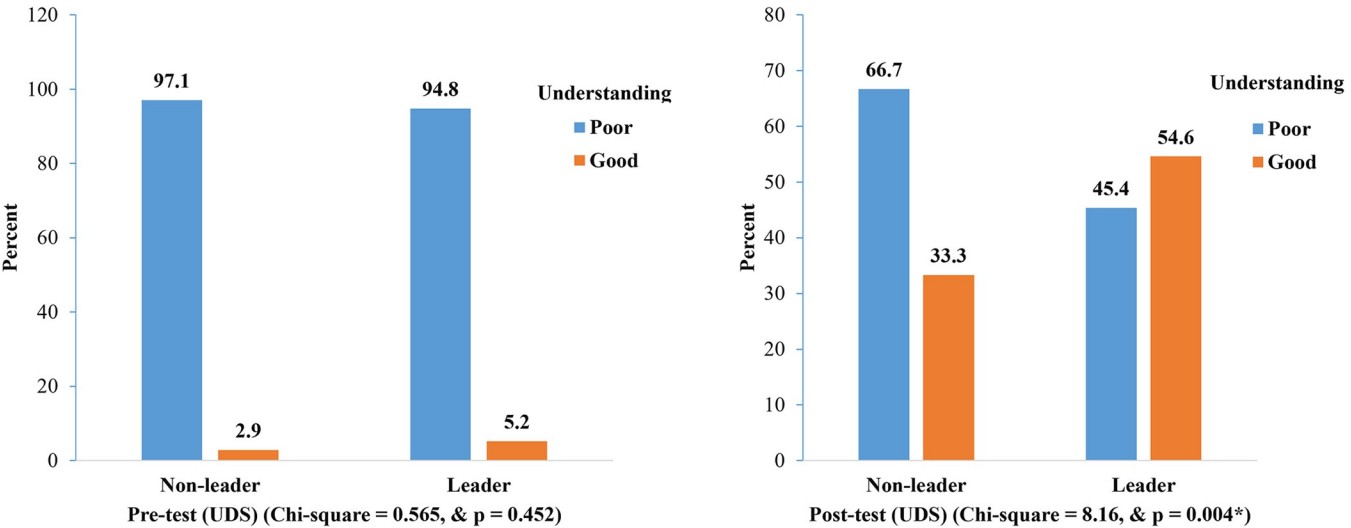

**Fig 1. Comparison of non-leader and leader students' understanding of dengue solutions (UDS).**

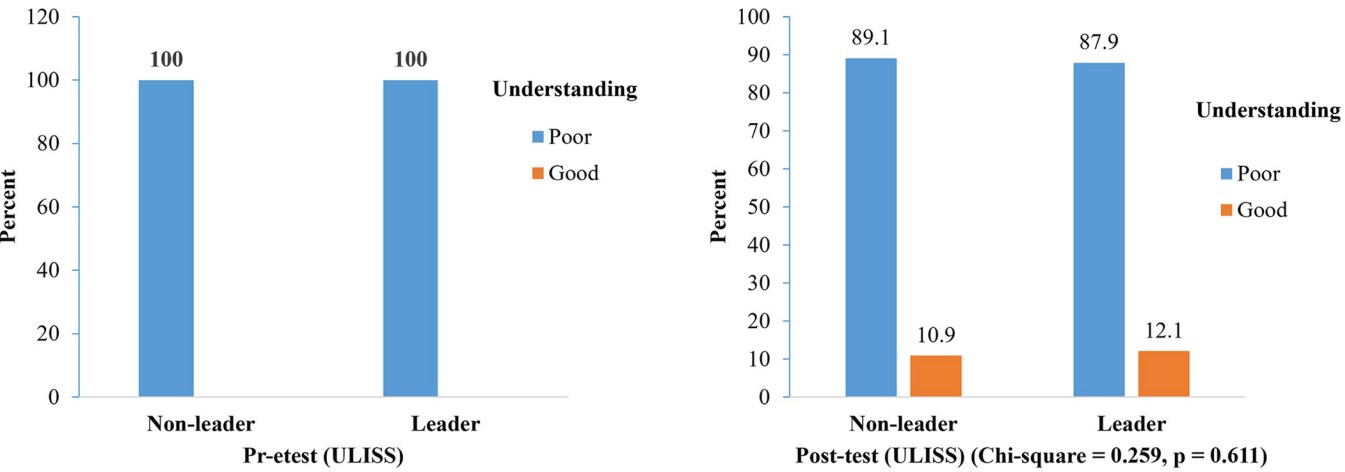

**Fig 2. Comparison of non-leader and leader students about their understanding of the larval indices surveillance system (ULISS).**

[27–30]. These children's age groups do attend school, so it is possible that they suffered significantly from a lack of understanding about prevention [31]. These children are more vulnerable to Aedes mosquito attacks because they play in open areas and close to the source of standing water. This can be explained by the Aedes mosquito's diurnal adaptation to stagnant or stored water, which occasionally attracts children [32]. The results demonstrated that a small percentage of leader and non-leader students have had dengue fever during the previous 12 months, while some students have observed the illness from other students, neighbours, or family members. Following the school-based dengue training, some students' experience status about the dengue illness changed from "yes" to "no" (or vice versa). There was a notable increase of over 94% in leader students who identified their neighbour as experiencing dengue sickness. This implied that some of the students could now distinguish dengue fever from other illnesses through the training intervention. It should be noted that malaria, typhus fever, enteric fever, leptospirosis, and other frequent tropical diseases are similar to dengue fever [33]. For example, Mayxay et al. [34] found that about one-third of their interviewees could

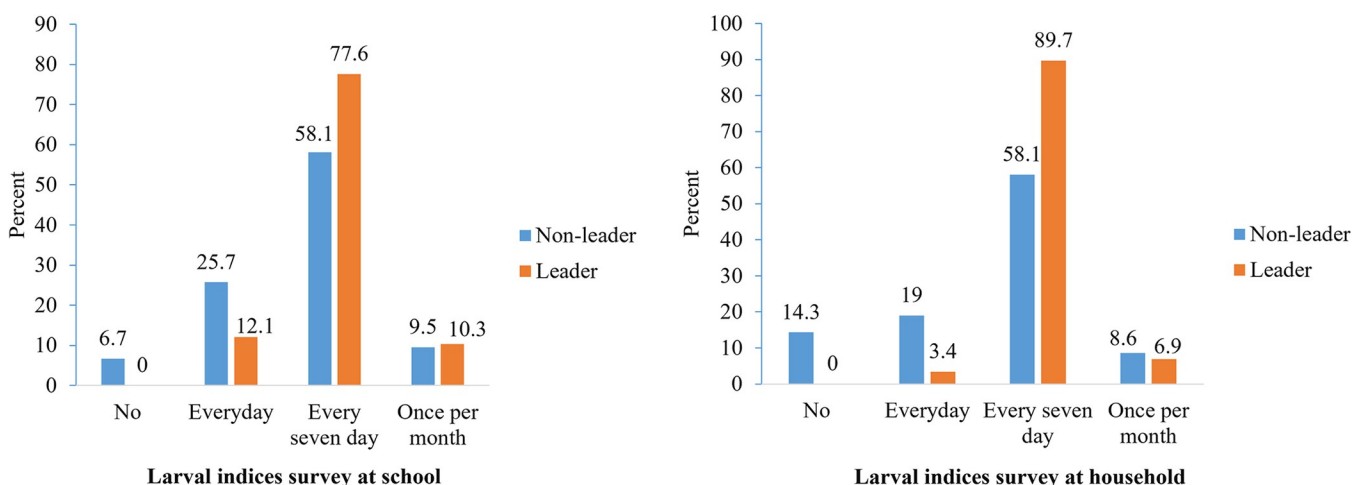

**Fig 3. Larval indices survey behaviour between non-leader and leader students.**

**Table 6. Thematic analysis of the student's responses.**

| Theme | Meaning | Response |
|---|---|---|
| **Knowledge acquisition and application** | This indicates that students learned about several elements of dengue mosquitoes, such as recognising breeding areas, calculating the Container Index (CI), and comprehending the life cycle. The practical application of information in decreasing mosquito larvae using safe ways and modifying the environment to avoid mosquito reproduction. | *"I can able to identify breeding grounds for dengue mosquitoes."* A2 <br> *"Learning that acquired knowledge can be used personally and shared with others."* A6 <br> *"Learning and answering questions about dengue fever."* A8 <br> *"I learned how to eliminate mosquito larvae, prevent dengue mosquitoes, and treat dengue fever."* A10 <br> *"Linking the transmission of dengue fever from mosquitoes to humans."* B10 <br> *"Learning how to inspect mosquito larvae, observing one's dengue fever symptoms, and those of friends."* C5 <br> *"Preventing and monitoring dengue mosquitoes; the life cycle of dengue mosquitoes."* D10 <br> *"Methods to protect against dengue mosquitoes, the knowledge that dengue mosquitoes are frightening, the danger of dengue fever. Dengue mosquitoes are active during the day and feed on human blood."* C3 |
| **Prevention and Self-Protection** | This indicates that students learned about mosquito bite prevention, dengue fever symptoms, and the necessity of self-care throughout the illness. The need to reduce mosquito breeding sites and inhibit mosquito larvae development was repeatedly underlined. | *"Eliminating mosquito larvae".* A2 <br> *"Destroying mosquito breeding grounds, eliminating mosquitoes from water sources."* A12 <br> *"Destroying mosquito egg-laying sites."* C12. <br> *"Changing the surrounding environment to prevent mosquito breeding, and preventing mosquito bites can be applied to every family member."* D14 |
| **Monitoring System** | It indicates that the students comprehend the Container Index (CI), a repeating concept, with participants learning about its many sorts and importance in managing dengue fever difficulties. | *"Solving dengue fever problems using the Container Index monitoring system."* B15. <br> *"Preventing dengue fever, preventing mosquitoes, the duration of dengue fever, and the CI monitoring system."* D9 |
| **Awareness and Understanding** | It strongly emphasises raising awareness of dengue mosquito prevention, understanding the hazards connected with these mosquitoes, and learning about their behaviours and traits. | *"The origin of dengue mosquitoes, why mosquitoes prefer biting overweight individuals, and understanding calculating the CI."* A2 <br> *"Realising the danger of dengue fever when bitten by dengue mosquitoes, understanding that dengue is more dangerous than initially thought."* A5 <br> *"Diseases from dengue mosquitoes are more severe than anticipated."* A7 <br> *"The danger of dengue fever. Dengue mosquitoes are active during the day and feed on human blood."* C3 <br> *"Understanding how dangerous dengue mosquitoes are, having fun, being with everyone during work."* C4 <br> *"Understanding which type of mosquito carries which disease and prevention methods."* C9 <br> *"Understanding the symptoms of dengue fever and the incubation period of larvae."* C12 <br> *"Understanding female dengue mosquitoes, and knowing what male and female dengue mosquitoes eat."* D1 <br> *"Learning about the danger of dengue fever, measuring the CI, and understanding which type of dengue mosquito causes dengue fever."* D8 <br> *"Understanding that even one dengue mosquito can harm a person."* D14 <br> *"Knowing that mosquitoes have various types and understanding the habitats of dengue mosquitoes."* D15 |

not distinguish malaria from dengue and misunderstood that Aedes mosquitoes transmit malaria. Javed et al. [35] state that a lack of understanding of dengue fever's symptoms can cause people to misidentify the disease for other febrile illnesses.

According to the results, after receiving training, leader students that have a good comprehension of dengue solution (UDS) increased from 5.2% to 54.6%. The results of Suwanbamrung et al. [36], which demonstrated that the student's basic knowledge scores were significantly higher after the intervention than before, are consistent with these findings about the intervention. Significantly poor knowledge prior to training intervention observed in this study was similar to the finding of Suwanbamrung et al. [18], who studied in 50 public primary schools in Kanchanadit district, Surat Thani province, where most students had limited knowledge of dengue prevention despite living in a dengue-endemic area. Despite an increase in understanding of the larval indices surveillance system (ULISS) among students from 100% poor understanding to 10.9–12.1% good understanding after training, there was still a prevalent lack of tangible understanding among more than 85% of the students. This was a very low grade for adequate comprehension compared to the students' good scores on UDS. Despite the training intervention, the students still do not fully understand how the larval indices surveillance system functions.

Similarly, Javed et al. [35] found that most children in their studied schools knew little or nothing about dengue illness, even after numerous awareness programs. However, Suwanbamrung et al. [37] noted that village health volunteers (VHVs) experienced a similar thing, as they could not compute the index values and were forced to recommend the larval index surveying training. In Thailand, the larval index survey was the most commonly used vector monitoring approach, and it was discovered to be an important active strategy for community-based dengue solutions [38,39]. Therefore, a lack of comprehensive understanding and implementation of vector control initiatives may diminish their overall efficacy and heighten the risk of the transmission of vector-borne diseases. The only way to stop the disease from spreading is to prevent it from happening in the first place, as there is no specific cure for it. Because dengue transmission is complicated and Aedes mosquitoes have distinct traits, vector management is essential. The goal of vector management measures is to interfere with the mosquito's life cycle in order to lower its population and the risk of dengue transmission [40].

This study shows that only one class and school among the leaders had the highest number of students who had a good comprehension of the larval indices surveillance system. Gender was also found to influence students' grasp of the larval indices surveillance system, with male students showing much better understanding than female students. According to Aronsi [41] and Byers et al. [42], a student's learning outcome is influenced by their class, gender, and learning environment. Though it was not statistically significant, females in UDS demonstrated better knowledge than their male counterparts. Similar results were also observed by Javed et al. [35], who found that female participants had a higher total score in signs and symptoms of the dengue domain and a significantly higher total score in knowledge of preventing the dengue domain, while male participants had a significantly higher total score in transmission of the dengue domain.

Regarding how often students practice larval indices surveillance, most leader students (89.7%) examined larval metrics at home and school every seven days. A small percentage of the leaders acknowledged engaging in larval indices surveillance daily or once a month, but none of them declined. Similar to that, Aung et al. [43] observed that a significant number of their studied students underwent self-reported experimental training at home to practice larval indices observation and efficiently explain it to family members. Also, more than half of the non-leader students practiced 'every seven days' larval indices surveillance at school and home. Because *Aedes aegypti* eggs take 7–10 days to develop into adult mosquitoes [44], it is advised to conduct a seven-day survey to remove vector habitat. However, given the poor ULISS scores of the students under study, it is likely that their procedures for surveying and calculating larval indices were inadequate. Even still, the majority of students said in the

study's thematic analysis that one of the skills they learned through the training was calculating larval indices. Although this study was conducted for 12 weeks, Khun and Manderson et al. [45] advocated a long-term school-based dengue program to ensure that knowledge is sustainable enough to put into action. Nevertheless, the only way to stop the spread of the dengue virus is to disable vector mosquito breeding places, and the general population must have adequate knowledge, attitudes, and good practices. According to Alobuia et al. [46], knowledge and attitude scores strongly predict practice scores. This suggests that those with superior knowledge and attitudes were more likely to take preventive steps to avoid vector infection.

Overall, it was discovered that the leader students could convey their knowledge of the dengue solution and the larval indices surveillance system to their peers who were non-leader students in the study. School-based health education plays a vital role in increasing awareness and knowledge about the severity of dengue among schoolchildren. It also facilitates the transfer of knowledge and practices from classrooms to homes. This is particularly important because dengue is common among schoolchildren, and efforts to control the disease's vectors require ongoing measures to reduce mosquito larval habitats [47]. A study conducted in Thailand found that implementing a program in primary schools increased children's understanding of and involvement in dengue prevention and control. This decreased the number of mosquito larvae found in schools' environments and the students' homes [48]. According to Ghasemi et al. [49], peer education is a health education approach in which individuals share knowledge, viewpoints, or abilities for various reasons, such as comparable age, sex, living conditions, and experiences, as well as cultural and social contexts. Moreover, peer education is a well-established concept that has been implemented in modern contexts. Students teaching students has been a cornerstone of Western pedagogy since at least the beginning of time in Greece [50], and the importance of cohort members as role models is widely acknowledged. Beyond educational roles, peers who engage others in participatory learning processes might help them feel empowered [51]. Likewise, teachers often have a significant impact on their students' lives. Thus, they play an important role in continuing to educate children about illness prevention and control. In the study of Suwanbamrung et al. [18], more than 70% of participants stated that teachers were their primary source of information on dengue infection, followed by parents and television.

Based on the thematic analysis, it was observed that the students had learned about dengue fever transmission, dengue mosquitoes, CI calculation, dengue symptoms, and how to prevent dengue mosquitoes, among other topics. Many students are aware that removing mosquito breeding grounds and avoiding mosquito bites is crucial for personal defence. Students were of the opinion that dengue mosquito-borne illnesses are more serious than they had thought. They knew the risk of dengue fever and that the mosquitoes feed on human blood and are active during the day. One student said that a single dengue mosquito can harm a person, while another mentioned knowing what both male and female dengue mosquitoes feed on. The primary vector for the spread of the dengue virus is the female *Aedes aegypti* mosquito species. According to Rund et al. [32], this particular mosquito species prefers to bite during the day, especially in the early morning and evening hours, just before dusk. This study has really informed the students and the schools about dengue infection and its solutions compared to what they knew before. Similarly, Aung et al. [43] documented that following an intervention training program, high school students in the Yangon region of Myanmar had a better understanding and a different perspective on dengue infection and its vector.

## Limitations

This study's limitations include the number of students participating and the disadvantages of purposive sampling. Some students who began with the pre-test study could not finish the

training or post-test study due to absences from school, reducing the number of recruited students. Another constraint encountered during the development of the training program was the language barrier, which was mitigated by delivering the program in both English and Thai, thus facilitating effective communication. Because of the semester school timetable and to prevent interruptions caused by semester holidays, the study was completed in 12 weeks. Meanwhile, we recommend a longer time frame for future studies so as to determine long-term outcomes.

## Conclusion

This study found that school-based dengue interventions are helpful in raising awareness and comprehension of dengue solutions. This study also implies that engaging elementary school students as health ambassadors to raise knowledge of personal preventative measures against dengue infection is a promising strategy for combating the disease. It has a significant impact on promoting community knowledge and compliance with vector management. The COVID-19 pandemic and the resulting lockdown measures have several adverse effects on dengue infection. These include a negative impact on preventing and controlling dengue strategies, surveillance and control interventions, and other issues. Given the students' low ULISS scores, we recommend that further ULISS training be provided for schoolchildren in simple terms that they can understand. To improve pupils' compliance with dengue prevention practices and the model's influence in the community, we encourage that their parents engage in future training sessions. Furthermore, sub-district health officials and health teachers should educate children, parents, and the wider community about dengue. Create a campaign to enhance community awareness of dengue control, hold regular town hall meetings, and distribute printed materials that can enlighten the public about dengue.

## Supporting information

**S1 Dataset. Microsoft excel file with minimal dataset.**
(XLS)

**S1 File. Descriptive findings of the study.**
(DOCX)

## Acknowledgments

We would like to acknowledge all participants in Keawsan subdistrict, including the SAO officials, PCUs, the households, family leaders, VHVs, CDCs, community leaders, teacher, student and Nakbon district public health officials.

## Author Contributions

**Conceptualization:** Jiraporn Jaroenpool, Muhammad Haroon Stanikzai, Chumpon Ponprasert, Temesgen Anjulo Ageru, Charuai Suwanbamrung.

**Data curation:** Jiraporn Jaroenpool, Chumpon Ponprasert, Shamarina Shohaimi, Temesgen Anjulo Ageru, Amin Nordin Syafinaz, Charuai Suwanbamrung.

**Formal analysis:** Jiraporn Jaroenpool, Sarunya Maneerattanasak, Atchara Phumee, Muhammad Haroon Stanikzai, Shamarina Shohaimi, Truong Thanh Nam, Temesgen Anjulo Ageru, Amin Nordin Syafinaz, Charuai Suwanbamrung.

**Funding acquisition:** Jiraporn Jaroenpool, Femi Adesina, Charuai Suwanbamrung.

**Investigation:** Jiraporn Jaroenpool, Atchara Phumee, Charuai Suwanbamrung.

**Methodology:** Jiraporn Jaroenpool, Femi Adesina, Muhammad Haroon Stanikzai.

**Project administration:** Atchara Phumee, Charuai Suwanbamrung.

**Resources:** Muhammad Haroon Stanikzai.

**Software:** Muhammad Haroon Stanikzai.

**Supervision:** Charuai Suwanbamrung.

**Validation:** Muhammad Haroon Stanikzai.

**Visualization:** Muhammad Haroon Stanikzai.

**Writing – original draft:** Jiraporn Jaroenpool, Sarunya Maneerattanasak, Femi Adesina, Muhammad Haroon Stanikzai, Yingqin Zheng, Shamarina Shohaimi, Truong Thanh Nam, Temesgen Anjulo Ageru, Amin Nordin Syafinaz, Charuai Suwanbamrung.

**Writing – review & editing:** Jiraporn Jaroenpool, Sarunya Maneerattanasak, Femi Adesina, Atchara Phumee, Muhammad Haroon Stanikzai, Chumpon Ponprasert, Yingqin Zheng, Shamarina Shohaimi, Truong Thanh Nam, Temesgen Anjulo Ageru, Amin Nordin Syafinaz, Charuai Suwanbamrung.

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
