## [Decision Letter · Decision Letter 0]

24 Sep 2024

PONE-D-24-29223A primary school-based dengue solution model for post-COVID-19 in southern Thailand: Students understanding of the dengue solution and larval indices surveillance systemPLOS ONE

Dear Dr. Suwanbamrung,

Thank you for submitting your manuscript to PLOS ONE. After careful consideration, we feel that it has merit but does not fully meet PLOS ONE’s publication criteria as it currently stands. Therefore, we invite you to submit a revised version of the manuscript that addresses the points raised during the review process.

We look forward to receiving your revised manuscript.

Kind regards,

Marianne Clemence, Staff Editor, on behalf of,

Rajib Chowdhury, M.Sc.; MPH

Academic Editor

PLOS ONE

Journal Requirements:

2. Thank you for stating the following financial disclosure: The project was funded by the Excellent Centre for Dengue and Community Public Health (EC for DACH) and School of Public Health, Walailak University.

3. Thank you for stating the following in the Acknowledgments Section of your manuscript: We would like to acknowledge all participants in Keawsan subdistrict, including the SAO officials, PCUs, the households, family leaders, VHVs, CDCs, community leaders, teacher, student and Nabon district public health officials. This work was supported by the Community

Public Health Program at the School of Public Health, Walailak University. This research was financially supported by EC for DACH, School of Public Health, Walailak University, and Keawsan subdistrict SAO, Thailand. They did not have any role in the study design, data collection analysis and interpretation, or in writing the manuscript.

The project was funded by the Excellent Centre for Dengue and Community Public Health (EC for DACH) and School of Public Health, Walailak University.

4. In the online submission form, you indicated that data will be made available on request.

Reviewers' comments:

Reviewer's Responses to Questions

**Comments to the Author**

1. Is the manuscript technically sound, and do the data support the conclusions?

Reviewer #1: Yes

Reviewer #2: Yes

Reviewer #3: Yes

2. Has the statistical analysis been performed appropriately and rigorously? 

Reviewer #1: I Don't Know

Reviewer #2: Yes

Reviewer #3: Yes

3. Have the authors made all data underlying the findings in their manuscript fully available?

Reviewer #1: Yes

Reviewer #2: Yes

Reviewer #3: No

4. Is the manuscript presented in an intelligible fashion and written in standard English?

Reviewer #1: Yes

Reviewer #2: No

Reviewer #3: Yes

5. Review Comments to the Author

Reviewer #1: The weakness of the article is sampling. purposive based on what inclusion criteria ? Why gender is significant different , is their any explanation or biasness. Authors can provide the tools as a supplement materials . Covid lockdown result in bad impact on dengue control , is it evidence enough by your study findings .

Reviewer #2: The research paper titled "A Primary School-Based Dengue Solution Model for Post-COVID-19 in Southern Thailand: Students' Understanding of the Dengue Solution and Larval Indices Surveillance System," by Jiraporn Jaroenpoola and colleagues, presents an intriguing and valuable model for preparedness and response in unprecedented situations. This model could be useful if it could be effectively replicated in other regions.

I have the following suggestions for enhancing the manuscript:

Abstract:

The abstract should concisely summarize the manuscript’s key findings, presenting them clearly and comprehensively. Revise the abstract by highlighting the most significant outcomes and selecting the most relevant keywords. Consider removing "Aedes aegypti" and instead focus on keywords that directly reflect the main findings and themes of the study.

Background:

Introduction to Dengue Virus Infection: A brief introduction to dengue virus infection outlines its significance as a public health concern, particularly in tropical and subtropical regions like Southern Thailand.

Current Situation of Dengue Virus Infection: Discuss dengue infections in Thailand overall and Southern Thailand, including any recent trends or data post-COVID-19. Highlight the challenges that have emerged due to the pandemic's impact on public health systems.

Need for a Dengue Solution Model: Explain why there is an urgent need for an innovative dengue solution model over another existing model, particularly in the context of primary schools. I suggest emphasising the importance of larval indices surveillance as a key component of this model and its potential to mitigate dengue transmission in the community.

Justification and Knowledge Gaps: I suggest providing a rationale for the current study by identifying gaps in contrast to existing research and knowledge. Justify the need for this research by explaining how it addresses these gaps and contributes to the understanding and control of dengue in post-COVID-19 settings.

Is there a decreased number of dengue cases in ASEAN countries directly linked with the movement of people or the reporting system since Singapore did not experience as with other ASEAN countries?

Methods:

Selection of Primary Schools: Justify the selection of the four primary schools included in the study. Provide criteria or reasoning that guided this selection, such as the geographical location, dengue risk levels, or previous history of dengue outbreaks. How do the investigators select leader and non-leader groups?

Pre-testing of the Developed Questionnaire: Elaborate on the pre-testing process of the developed questionnaire. Explain how this process ensured the reliability and validity of the data collection tool, and how it was tailored to the context of the study.

Integration of the Dengue Model: Describe how the dengue solution model was integrated into the school-based setting. Discuss the steps taken to ensure that the model was both practical and adaptable to the needs of the students and the school environment.

Assurance of Model Effectiveness: Outline the methods used to evaluate and validate the effectiveness of the developed and adapted dengue control model. Provide details on how the model’s success was measured and any criteria used to assess its impact.

Evaluation of the Dengue Control Model: Clarify whether the study evaluated the dengue control modelling method and, if so, describe the evaluation process in detail, including the tools and indicators used to measure its success.

Results:

Presentation of Findings: Construct a 2x2 table to present the major findings of the study in a clear and concise format and tables Summarize key outcomes, focusing on the most significant results rather than getting bogged down in excessive detail.

What was the impact of the study on the parents and the family members?

Discussion:

Is there any previous study Comparison with Previous Studies: Compare the findings of this study with previous research conducted on similar topics. Highlight any similarities or differences and provide logical explanations for any discrepancies observed between this study and existing literature.

Limitations:

Study Limitations: Discuss the limitations of the study, including any constraints or potential biases that could affect the findings. Be transparent about these limitations to provide a balanced view of the study's strengths and weaknesses.

Suggestions for Future Research: Offer suggestions for how these limitations could be addressed in future research. Identify areas where further investigation is needed to build on the findings of this study. Is there a further expansion of the study in other areas?

Implementation Challenges: Address any potential barriers to implementing the recommendations from this study. Propose feasible and actionable solutions to overcome these challenges, ensuring that the findings can be effectively translated into practice.

Recommendations: It would be very useful if the authors could provide recommendations to the community, the municipal government and relevant stakeholders.

Reviewer #3: The study "A primary school-based dengue solution model for post-COVID-19 in southern Thailand: Students understanding of the dengue solution and larval indices surveillance systemis" is the result of rigorous research that provides useful information for those working in the education and prevention of dengue and other communicable diseases. The researchers applied appropriate methodologies to achieve the objectives, as it is a mixed-methods study with both qualitative and quantitative methodologies. The analyses are detailed and apply relevant statistics to each case, and the results are correctly expressed. The figures and tables provide sufficient and easy-to-understand information. The sample size is adequate, and although some participants dropped out of the study, the analysis was still completed. The results adequately characterize the study population and provide evidence for the conclusions.

The statistical analysis is the best developed part of the manuscript.

The researchers propose an interesting model that is valid for dengue prevention in primary school children, however it would be useful if they described the limitations they found in developing the training of children and the recommendations to increase the impact of the model or the adjustments they would suggest to achieve percentages higher than 80% in all aspects of dengue prevention.

Most of the references are appropriate, but I have pointed out three changes in the citation in the text (see annex)

Apply reference to this statement from the introduction section: The movement of individuals over national borders is the main factor contributing to the spread of both COVID-19 and dengue infection, which increases the likelihood of a continuous outbreak.

And

In the first paragraph of the discussion: This can be explained by the Aedes mosquito's diurnal adaptation to stagnant or stored water, which occasionally attracts children.

Find the reference to the vector biology study, which determines the diurnal biting behavior for Aedes, and therefore change citation number 26 (N. Javed, H. Ghazanfar, S. Naseem. Knowledge of Dengue Among Students Exposed to Various Awareness Campaigns in Model Schools of Islamabad: A Cross-Sectional Study. Cureus 10 (2018) e2455. DOI 10.7759/cureus.245)

6. PLOS authors have the option to publish the peer review history of their article (what does this mean?). If published, this will include your full peer review and any attached files.

Reviewer #1: **Yes: **Ariful Basher

Reviewer #2: No

Reviewer #3: No

---

## [Author Response · Author response to Decision Letter 0]

7 Oct 2024

Thank you. We have uploaded a detail response letter.

---

## [Editor Report · Decision Letter 1]

21 Oct 2024

A primary school-based dengue solution model for post-COVID-19 in southern Thailand: Students understanding of the dengue solution and larval indices surveillance system

PONE-D-24-29223R1

Dear Dr. Suwanbamrung,

We’re pleased to inform you that your manuscript has been judged scientifically suitable for publication and will be formally accepted for publication once it meets all outstanding technical requirements.

Kind regards,

Rajib Chowdhury, M.Sc.; MPH

Academic Editor

PLOS ONE
---

## [Editor Report · Acceptance letter]

24 Oct 2024

PONE-D-24-29223R1 

PLOS ONE

Dear Dr. Suwanbamrung, 

I'm pleased to inform you that your manuscript has been deemed suitable for publication in PLOS ONE. Congratulations! Your manuscript is now being handed over to our production team.

Kind regards, 

on behalf of

Dr. Rajib Chowdhury 

Academic Editor

PLOS ONE